# Characterization of Sodium Alginate-Based Films Blended with Olive Leaf and Laurel Leaf Extracts Obtained by Ultrasound-Assisted Technology

**DOI:** 10.3390/foods12224076

**Published:** 2023-11-09

**Authors:** Márcio Moura-Alves, Victor Gomes Lauriano Souza, Jose A. Silva, Alexandra Esteves, Lorenzo M. Pastrana, Cristina Saraiva, Miguel A. Cerqueira

**Affiliations:** 1CECAV—Centre for Studies in Animal and Veterinary Science, University of Trás-Os-Montes and Alto Douro, 5000-801 Vila Real, Portugal; jasilva@utad.pt (J.A.S.); alexe@utad.pt (A.E.); crisarai@utad.pt (C.S.); 2AL4AnimalS—Associate Laboratory for Animal and Veterinary Sciences, 5000-801 Vila Real, Portugal; 3INL—International Iberian Nanotechnology Laboratory, Av. Mestre José Veiga s/n, 4715-330 Braga, Portugal; victor.souza@inl.int (V.G.L.S.); lorenzo.pastrana@inl.int (L.M.P.); miguel.cerqueira@inl.int (M.A.C.)

**Keywords:** edible films, plant extracts, antimicrobial activity, antioxidant activity, barrier properties, mechanical properties, physico-chemical properties, active films

## Abstract

Due to environmental concerns, there is an increasing need to reduce the use of synthetic and non-renewable packaging materials to reduce waste and increase sustainability. This study aimed to characterise sodium alginate edible-based films (SA) incorporated with laurel leaf extract (LLE) and olive leaf extract (OLE) obtained by ultrasound-assisted extraction. Determination of total phenolic content, antioxidant, and antimicrobial activity was performed for the extracts and films. Also, thickness, tensile strength, elongation at break, modulus of elasticity, opacity and colour, moisture content, water vapour permeability (WVP), Fourier-transform infrared spectroscopy (FTIR) spectra, and surface morphology by scanning electron microscope (SEM) analyses were performed for the films. LLE yielded better results in terms of phenolic content (195 mg GAE/g), antioxidant (2.1 TE/g extract) and antimicrobial activity (MIC at 1% for *Listeria monocytogenes* and *Staphylococcus aureus*, and 1.8% for *Enterococcus faecalis*). For the films, the simultaneous incorporation of LLE 1% (*w*/*v*) and OLE 1% (*w*/*v*) resulted in a significant reduction of approximately 2 log CFU/g against *S. aureus*. The addition of LLE and OLE extracts also proved to improve barrier properties (lower WVP for SA films with LLE 1% + OLE 1%, 3.49 × 10^−11^ g m^−1^ s^−1^ Pa^−1^) and promoted changes in resistance and flexibility. The results demonstrated that active alginate-based films can be valuable for enhancing food preservation.

## 1. Introduction

Food packaging plays a crucial role in food protection and preservation. Petroleum-based packaging materials constitute one of the most used products. However, they are also well-known for their significant contribution to serious environmental problems [1]. To address these environmental concerns, researchers have directed their efforts toward the development of ecological packaging solutions obtained from renewable resources to minimise the negative impact on our planet [2,3]. Natural polymers such as proteins, polysaccharides, lipids, and their combinations have been exploited for this purpose due to their edibility, biodegradability, biocompatibility, non-toxicity, and the potential to provide antioxidant and antimicrobial activity [4,5].

The use of alginate-based films in food applications has increased in recent years. Alginates are hydrophilic colloidal carbohydrates obtained from marine brown seaweed with a linear structure primarily composed of (1,4)-linked *β*-_D_-mannuronic acid units and *α*-_L_-guluronic acid units [6]. They are generally recognised as safe (GRAS) according to the Food and Drug Administration (FDA) [7]. Additionally, the European Commission (EC) has officially authorised alginic acid and its salts (E400–E404) as food additives [8]. Several studies have shown the potential to extend the shelf-life of food products by combining alginates and plant-derived compounds such as pure bioactive compounds, extracts of fruits, roots, leaves, seeds, their essential oils, peptides, etc., due to their antimicrobial, antioxidant, and anti-browning properties [3,9]. It was also reported that the presence of phenolic compounds can modify the physico-chemical, mechanical, and barrier properties of films, making it possible to improve them [3,10]. Plant-based bioactive compounds in combination with alginate have been tested for food preservation. The addition of *Vitis vinifera* grape extract to alginate-base films improved their structural, antimicrobial, and antioxidant activities [10]. In another study, the shelf-life of tomatoes was extended by coating them with alginate films incorporated with both aloe vera and garlic oil [11]. Raeisi et al. [12] also proved positive outcomes in preserving chicken meat using alginate films containing cinnamon and rosemary. The antimicrobial compounds exhibited a stronger effect when combined rather than individually [12]. Many bioactive compounds and combinations are yet to be explored regarding their bioactive properties, physico-chemical characteristics, and their applicability.

*Laurus nobilis* L., belonging to the *Lauraceae* family, commonly known as bay laurel or sweet bay, is an aromatic evergreen native to the Mediterranean region. Its leaves have traditionally been used in culinary, folk medicine, and ornamental applications [13]. The extract derived from the leaves has been studied using different extraction methods and solvents, showing promising results due to their antioxidant and antimicrobial properties. This activity is primarily attributed to phenolic compounds present in *L. nobilis* leaves including flavonoids, phenolic acids, tannins (proanthocyanidins), and lignans [13,14,15].

Olive (*Olea europaea* L.) is one of the most globally cultivated ancient plants, representing a fundamental cultivar of the Mediterranean region [16]. The cultivation and processing of olives generate a significant volume of by-products and residues annually [17]. The leaves are found in substantial amounts during tree pruning and harvesting, typically being discarded or burned [17]. Nevertheless, they have a high potential for additional value by recovering valuable bioactive compounds suitable for food and pharmaceutical products [18]. Extensive research has focused on their phenolic compounds due to their numerous beneficial effects on human health. The main polyphenols present in olive leaves are divided into five groups: oleuropeosides (oleuropein and verbascoside), flavones (luteolin-7-glucoside, apigenin-7-glucoside, diosmetin-7-glucoside, luteolin and diosmetin), flavonols (rutin), flavan-3-ols (catechin), and substituted phenols (tyrosol, hydroxytyrosol, vanillin, vanillic acid and caffeic acid) [19]. Oleuropein and hydroxytyrosol are reported as the most abundant compounds followed by flavone-7-glucosides of luteolin, apigenin, and verbascoside [19].

Traditionally, the extraction of bioactive compounds from plants has been accomplished through maceration, using liquid solvents [20]. However, this method is considered time-consuming due to its extended extraction times [20,21]. Ultrasound-assisted extraction (UAE) is considered one of the most interesting sustainable alternatives [21,22]. UAE is based on the principle of acoustic cavitation where sound waves are applied to disrupt plant cell walls, facilitating the release of bioactive compounds [23,24]. This technology is easy to handle, safe, cost-effective, reproducible, and can improve the extraction yield of compounds compared to conventional techniques [22,23].

Given the existing challenges in the food industry, this study aimed to develop an active edible film using sustainable materials and technologies. Sodium alginate was chosen as the polymeric matrix and the leaves of two species with growth in Portugal and the Mediterranean Region, laurel and olive, were chosen as the bioactive sources known for their antioxidant and antimicrobial potential. There are numerous studies that have investigated the properties and effects of individual plant extracts [2,25]. However, research on the combined use of multiple plant extracts is typically less common. The combination of laurel and olive leaf extracts has not been tested before and the combined effects of different plant extracts can provide unique insights, as these combinations can have synergistic or additive effects. Sodium alginate (SA) films incorporated with leaf extracts of laurel (LLE) and olive (OLE) were produced and fully characterised for their mechanical and physico-chemical properties, morphology, and antimicrobial and antioxidant activity.

## 2. Materials and Methods

### 2.1. Extraction Procedures

#### 2.1.1. Sample Preparation

Laurel leaves (*Laurus nobilis*) were collected in April from trees at the University of Trás-os-Montes e Alto Douro, Vila Real, Portugal (41.28763, −7.74043). Olive leaves (*Olea europaea*) were collected in May on a farm from Resende, Portugal (41.10300, −7.98156). The leaves were stored at 4 °C and processed in less than 48 h. The leaves were washed in running water and then in distilled water. The leaves were carefully sorted by hand, and any diseased leaves were removed from the batch. The leaves were dried at 25 °C in an oven equipped with a fan to promote air circulation (Memmert Incubator IF260, Memmert GmbH + Co. KG, Schwabach, Germany) for ~7 days (until constant weight) and milled with an Ultra Centrifugal Mill Type ZM200 (Retsch, Germany) equipped with a 1 mm sieve at 10,000 rpm.

#### 2.1.2. Ultrasound-Assisted Extraction (UAE)

Dried and powdered leaves (20 g) were extracted with a 70:30 (*v*/*v*) ethanol-distilled water solvent solution (100 mL) in an ultrasound bath filled with 3 L of distilled water at 37 kHz (Elmasonic S60, 150 W, Elma Schmidbauer, Germany) for 1 h at 25 °C ± 5 °C in the dark. During the extraction, the water bath temperature was continuously monitored to keep it at 25 °C. Crude extracts were centrifuged at 5000× *g* (Gyrozen model 1248 R, Incheon, Republic of Korea) for 10 min. The solvent was removed using a rotary evaporator (IKA RV 8 V, Staufen, Germany) at 38 °C under vacuum. Finally, the extracts were freeze-dried (Dura Dry TM μP freeze-drier; −45 °C) and stored at 4 °C. The extraction parameters were chosen based on current literature [26,27,28].

The extraction yield for each extract was calculated by equation (Equation (1)) where *W*_1_ is the mass of lyophilised extract, and *W*_2_ is the mass of dried matter used for the extraction.
(1)$$Extraction\;yield\;(\%)=\frac{W_{1}}{W_{2}}\times 100$$

### 2.2. Film Forming Solutions (FFS) and Film Preparation

The FFS was obtained by mixing 1% (*w*/*v*) of sodium alginate (SA) 90.8–106.0% (PanReac AppliChem, Darmstadt, Germany) and 0.5% (*w*/*v*) of glycerol 99.95% (José Manuel Gomes dos Santos, Odivelas, Portugal) in distilled water under agitation overnight at room temperature. The lyophilised extracts were dissolved in distilled water, stirred for 1 h, filtered under vacuum using a QNHS^−1^50-100 filter, pore size 7–9 µm (Prat Dumas, Couze-et-Saint-Front, France), and added to the film-forming solutions in the following concentrations: laurel leaves extract (LLE) at 1 and 2% (*w*/*v*), olive leaves extract (OLE) at 1 and 2% (*w*/*v*), and a mixture of LLE 0.5% (*w*/*v*) + OLE 0.5% (*w*/*v*), and LLE 1% (*w*/*v*) + OLE 1% (*w*/*v*). All solutions were stirred for 1 h. The dispersion of the extracts in the alginate matrix was optimised in the preliminary work, aiming for a good dispersion of the extracts. The use of mixtures of ethanol/water as solvent presented interesting dispersion capabilities, and the solutions were homogenized with an Ultra-Turrax (IKA T18 digital Ultra-Turrax, Staufen, Germany) at 10,000 rpm for 2 min and degassed under vacuum. The film-forming solutions were cast in polystyrene petri plates (28 mL to 96 mm petri plates), dried at 35 °C (with air circulation) for 24 h, and conditioned in desiccators containing a saturated solution of Mg(NO_3_)_2_·6H_2_O (Sigma Aldrich, Darmstadt, Germany) at 53% of relative humidity (RH) and at 20 °C before analysis. Tree replicates of each film were prepared for further analysis.

### 2.3. Determination of Total Phenolic Content (TPC)

The determination of TPC in extracts and films was determined by the Folin–Ciocalteu method [29]. The extracts were reconstituted in distilled water (1 mg/mL) and filtered using a 0.20 μm syringe filter. Briefly, 1 mL of extract or film diluted in distilled water was mixed with 0.5 mL of Folin-Ciocalteu reagent (PanReac Applichem, Darmstadt, Germany) and 4.5 mL of distilled water. After 5 min, 4 mL of Na_2_CO_3_ (Merck, Germany) 7.5% (*w*/*v*) solution was added to the mixture. The reaction was kept in the dark at room temperature for 2 h. Absorbance was measured using a UV-Vis spectrophotometer (Jasco V-530 UV/VIS, Tokyo, Japan) at 765 nm and the results were expressed as mg of gallic acid equivalent (GAE) per g of extract/film from the calibration curve. All experiments were carried out in triplicate.

### 2.4. Antioxidant Activity

The ABTS scavenging activity assay was conducted following the procedure described by Re et al. [30]. The ABTS radical cation solution was prepared by mixing (1:1) 7 mM of ABTS (2,20-Azino-bis(3-ethylbenzothiazoline-6-sulfonic acid)) diammonium salt (VWR, EUA) and 2.45 mM potassium persulphate (Merck, Darmstadt, Germany). The mixture was kept in the dark overnight for 12–16 h prior to use. The radical solution was then diluted with ethanol (Chem-Lab, Zedelgem, Belgium) to obtain an absorbance of 0.70 ± 0.02 at 734 nm.

For the assay, a mixture containing 2 mL of the diluted radical solution and 20 µL of the diluted extract (filtered using a 0.20 μm syringe filter), films, controls, or standards were prepared. The absorbance of this mixture was measured after 6 min at 734 nm. Trolox (Acros Organics, Geel, Belgium) was used as a standard solution at a concentration range between 252.5 and 2020 µM in ethanol (final concentration on cuvette). The results were presented as mmol Trolox equivalent (TE)/g of lyophilised extract.

The *IC*_50_ value was also determined for the extracts as the concentration of the compounds that resulted in a 50% inhibition of the radical scavenging activity (RSA) using Equation (2) where ${Abs}_{c}$ is the absorbance of the radical + water (blank) and ${Abs}_{s}$ is the absorbance of the radical with the extract. BHT (butylated hydroxytoluene) (Merck, Darmstadt, Germany) in ethanol (100%) was used for comparison. All measurements were performed in triplicate. The results were expressed as mg/mL.
(2)$$Percentage\;of\;inhibition=\frac{{Abs}_{c}-{Abs}_{s}}{{Abs}_{c}}\times 100$$

### 2.5. Antimicrobial Activity

The antimicrobial activity of the extracts was conducted following the broth microdilution method using an ELISA plate reader (BioTek model PowerWave XS2 Winooski, BioTek Instruments, Winooski, VT, USA) at 600 nm. A maximum of 2% (*w*/*v*) of extracts was used (filtered using a 0.20 μm syringe filter) and serial dilutions were made with brain heart infusion broth (BHI) (Liofilchem, Roseto degli Abruzzi, TE, Italy). *Listeria monocytogenes* ATCC 7973, *Staphylococcus aureus* ATCC 25923, *Enterococcus faecalis* ATCC 19433, *Salmonella* Typhimurium ATCC 14028, and *Escherichia coli* ATCC 11775, were tested at ~5 × 10^5^ CFU/mL, standardised by OD600. Minimal inhibitory concentration (MIC) values were determined as the lowest concentration of the extract corresponding to values of optical density (OD) comparable to those of cell-free BHI. The minimum bactericidal concentration (MBC) was determined by the subculture of each well with no visible growth on selective agar plates. Oxford Listeria agar (Liofilchem, Roseto degli Abruzzi, TE, Italy) was used for *L. monocytogenes* (incubated at 37 °C for 48 h), Baird-Parker agar (VWR, Leuven, Belgium) for *S. aureus* (incubated at 37 °C for 48 h), Slanetz agar (HiMedia, Maharashtra, India) for *Enterococcus faecalis* (incubated at 37 °C for 48 h), Hektoen Enteric agar (Liofilchem, Roseto degli Abruzzi, TE, Italy) for *Salmonella* Typhimurium (incubated at 37 °C for 48 h) and Tryptone Bile X-Glucuronide agar (Liofilchem, Roseto degli Abruzzi, TE, Italy) for *E. coli* (incubated at 40 °C for 48 h).

The antimicrobial activity of the films was determined by the viable cell count assay method according to Nouri et al. [31] with slight modifications. Samples with 0.1 g were immersed in 2 mL of BHI and inoculated with ~10^6^ CFU/mL of the microorganisms previously mentioned. All microorganism concentrations were standardised by OD600. Samples were incubated at 37 °C and counts were obtained at 0 and 24 h. From the test tubes, 1 mL of sample was taken and added with 9 mL Tryptone Salt (HiMedia, Maharashtra, India) for the 10-fold serial dilution and enumeration using the agar plate medium previously mentioned. Results were reported as CFU/g.

### 2.6. Migration of Bioactive Compounds

The migration assay was determined according to Mariño-Cortegoso et al. [32] with a few changes. Different simulants were used: water, 10% ethanol (*v*/*v*, in distilled water), and 95% ethanol (*v*/*v*, in distilled water). Square samples of 1 cm^2^ of each film were mixed with 1.67 mL of the simulant, achieving an area-to-volume ratio of 6 dm^2^/L. Samples were placed in an oven at 1, 3, 6, 9, and 15 days at 40 °C. After this period, the total content of phenolic compounds (Section 2.3) and the antioxidant capacity (Section 2.4) were determined.

### 2.7. Physico-Chemical Properties of Films

#### 2.7.1. Thickness

The film thickness (µm) was measured with an electronic micrometre (Schut, BN Groningen, The Netherlands). Five measurements were taken on each sample at different and random points. The mean values were used to calculate water vapour permeability (WVP).

#### 2.7.2. Water Vapour Permeability (WVP)

The measurement of WVP was performed gravimetrically based on ASTM E96/E96M-10 [33]. The payne permeability cups (Elcometer 5100, Nieuwegein, The Netherlands) were filled with anhydrous calcium chloride (Sigma Aldrich, Darmstadt, Germany) to generate an RH of ~0%. Then, the films were sealed on the top of these cups and placed in a desiccator containing saturated Mg(NO_3_)_2_·6H_2_O (Sigma Aldrich, Darmstadt, Germany) to maintain a 53% RH and equipped with a fan to promote air circulation. The cups were periodically weighed at intervals of 2 h for 12 h at ambient temperature. The slope of weight gain (g) versus time (s) was obtained by linear regression and the water vapour transmission rate (WVTR) was calculated from the slope of the straight line (g/s) divided by the cell area (m^2^). WVP (g m^−1^ Pa^−1^ s^−1^) was calculated by Equation (3) where e is the film thickness (m) and ∆P is the partial pressure difference across the film. Three replicates were obtained for each sample. Polylactic acid films (Goodfellow, Huntingdon, UK) with 0.05 mm of thickness were also tested for comparison. Temperature and humidity inside the desiccator were also monitored using a thermohygrometer (iButton DS1923, Newbury, UK).
(3)$$WVP=WVTR\times \frac{e}{∆P}$$

#### 2.7.3. Colour and Opacity

The colour of the films was determined using a Minolta colorimeter (CR 400; Minolta, Japan) calibrated by a standard white plate. CIELab parameters *L**, *a** and *b** of each film were recorded by reflectance measurements. The changes in the surface colour were measured by the Hunter total colour difference (Δ*E*) using Equation (4).
(4)$$\Delta E=\sqrt{{{(L}^{*}\;-\;L_{0}^{*})}^{2}+{{(a}^{*}\;-\;a_{0}^{*})}^{2}+{{(b}^{*}\;-\;b_{0}^{*})}^{2}\;}$$

The opacity of the samples was determined according to the Hunter lab method, relating the opacity of each sample on a black standard (*Y_b_*) and the opacity of each sample on a white standard (*Y_w_*) (Equation (5)). Five replicates were obtained for each sample film. Polylactic acid films (Goodfellow, Huntingdon, UK) were also tested for comparison.
(5)$$Opacity=\frac{Y_{b}}{Y_{w}}\times 100$$

#### 2.7.4. Moisture Content and Water Solubility

To determine the moisture content of films, approximately 1 cm^2^ square samples were cut and weighed and then subjected to drying at 105 °C for 24 h, allowing the samples to reach an equilibrium weight. The weight loss of the samples after drying was measured as a percentage according to Equation (6), where *Mi* and *Mf* are the masses of initial and dried samples, respectively. The analyses were performed in triplicate.
(6)$$Moisture\;content\;(\%)=\frac{Mi-Mf}{Mi}\times 100$$

To determine the film’s solubility in water, the dried films were immersed in 50 mL at 20 °C with agitation (150 rpm) for 24 h. The insoluble pieces of film were taken out and dried to a constant weight in an oven at 105 °C. The weight of dry matter that was not solubilised was measured and converted to water solubility (%) using Equation (7), where *Mi* and *Mf* are the masses of initial and dried samples, respectively.
(7)$$Water\;solubility\;(\%)=\frac{Mi-Mf}{Mi}\times 100$$

### 2.8. Mechanical Properties

The mechanical properties of the films were measured using an universal testing machine (AGX-V 10 kN, Shimadzu, Kyoto, Japan) equipped with a 500 N load cell according to ASTM D882-02 [34]. Two strips of each film (20 mm wide and 70 mm long) were mounted in the tensile grips with an initial gauge length of 50 mm and stretched at a cross-head speed of 50 mm/min until breakage. The tensile strength (expressed in MPa), Modulus of Elasticity (expressed in MPa) and elongation-at-break (expressed in %) were obtained from the stress-strain curve.

### 2.9. Fourier Transform Infrared Spectroscopy (FTIR)

The FTIR spectra of the films were analysed using attenuated total reflection mode (ATR) in an FTIR Spectrometer (Vertex 80v, Bruker, Mannheim, Germany). A wavelength range between 4000 and 600 cm^−1^ at a resolution of 4 cm^−1^ was used. An average of 64 scans were collected for each scan. The absorbance of each FTIR spectrum was normalised between 0 and 1.

### 2.10. Scanning Electron Microscopy (SEM)

The surface morphology of the films was examined using scanning electron microscopy (Quanta FEG 650, FEI, Hillsboro, OR, USA) with an accelerating voltage of 5 kV. Before analysis, all samples were mounted on aluminium stubs using carbon adhesive tape and sputter-coated with gold (thickness of about 10 nm) in a vacuum coater (Leica EM ACE200, Wetzlar, Germany).

### 2.11. Statistical Analysis

Results are presented as the mean ± standard deviation (SD). Statistical analysis was performed using the analysis of variance One-Way ANOVA. Comparison of means was achieved by a Tukey HSD test (“Honestly Significantly Different”), for a significance level of 5%. Data analyses were performed using the Statistica software version 12 program (StatSoft, 2014, Tulsa, OK, USA).

## 3. Results and Discussion

### 3.1. Extraction Yield of Extracts

For 1 kg of laurel leaves, 480 g of dried leaves were obtained, which resulted in 60.0 g of lyophilised extract. The corresponding yield was 6.0% from wet bases and 12.5% from dry leaves. Inés Molina et al. [35] obtained extract yields between 1 and 10.6% from *Laurus nobilis* dry leaves prepared by maceration using different solvents. The highest yields were obtained with methanol (10.6%). The results obtained in this study were higher which may be explained by the technology and solvent used.

For olive leaves, 1 kg resulted in 505 g of dried leaves and 72.1 g of extract. Thus, 7.2% and 14.3% of yield was obtained from wet and dry bases, respectively. The results are in agreement with other published works. Şahin and Şamlı [28] obtained 20.4% from dry leaves, using 50% ethanol (1:20, *w*/*v*) and 1 h of sonification in an ultrasound bath. Borges et al. [36] obtained an extraction efficiency of 4.6% (wet weight) for OLE using ethanol (1:10, *w*/*v*) and 1 h of sonification in an ultrasound bath. Bilgin and Şahin [37] reported extract yields between 8.9% and 35.1% from dry leaves using methanol (1:20, *w*/*v*) and 1 h in an ultrasound bath.

### 3.2. Total Phenolic Compounds and Antioxidant Activity of Extracts

The results for TPC and antioxidant activity of extracts and BHT are shown in Table 1. LLE showed the best results for phenolic compounds with 195.75 GAE/g of extract followed by LLE + OLE (179.13 GAE/g of extract) and OLE (147.04 GAE/g of extract).

In this study, the yield of phenolic compounds for LLE was 11.71 mg GAE/g from wet leaves and 24.40 mg GAE/g from dry leaves. The values were higher than other studies using the same technology. Muñiz-Márquez et al. [26] obtained 9.26 GAE/g for wet leaves, using 70% ethanol (1:4, *w*/*v*) and 1 h in an ultrasound bath. The same author reported an increase in yield to 14.69 GAE/g for wet leaves using 1:8 solid/liquid ratio instead of 1:4. In another study, Rincón et al. [38] obtained 7.03 GAE/g from dry leaves using an ultrasonic bath for 1 h and ethanol as a solvent 1:40 (*w*:*v*). The yield of phenolic compounds obtained for OLE was 10.60 mg GAE/g from wet leaves and 20.98 mg GAE/g from dry leaves. Şahin and Şamlı [28] obtained 19.51 GAE/g from dry leaves, using 50% ethanol (1:20, *w*/*v*), and 1 h in an ultrasound bath. Bilgin and Şahin [37] evaluated olive leaves from six different geographical origins in Turkey. The extraction was performed using methanol (1:20, *w*/*v*) and 1 h of sonification in an ultrasound bath. The total phenolic content varied from 7.35 to 38.66 mg GAE/g from dried leaves.

An order of LLE > LLE + OLE > OLE was also observed for antioxidant capacity. At 1 mg/mL in water, higher TEAC levels were obtained for LLE (2.07 mmol TE/g extract), and lower levels for OLE (0.78 mmol TE/g extract). For BHT, an antioxidant compound used as a preservative in foods, 1.53 mmol TE/g was obtained at the same concentration tested for the extracts. These values were lower than those obtained for LLE, which indicated a good antioxidant capacity of LLE at these concentrations. For IC_50_, the best antioxidant capacity was observed for BHT (0.56 mg/mL), followed by LLE (0.74 mg/mL), LLE + OLE (0.98 mg/mL), and OLE (1.77 mg/mL).

The results obtained are different from those obtained in different studies. Vallverdú-Queralt et al. [39] obtained 0.72 mmol TE/g from dry laurel leaves (50% ethanol (1:5, *w*/*v*) with 5 min of sonification). Oudjedi et al. [40], using a traditional extraction (without sonification), obtained an IC_50_ of 18.68 µg/mL and 14.65 µg/mL for LLE with 60% ethanol and 80% ethanol, respectively. For OLE, Ahmad-Qasem et al. [41] reported an extraction of 7.2 mg TE/g from dry leaves by applying an ultrasound probe system and 80:20 (*v*/*v*) during 15 min in an ethanol–water solution (1:32, *w*/*v*). Borges et al. [36] reported values of 209 TE/g from a fresh mass of olive leaves at 5 mg/mL of extract (ethanol (1:10, *w*/*v*) and 1 h in an ultrasound bath).

The differences in the results of extraction yields, phenolic compounds, and antioxidant activity can be explained by the variability related to the plant origin, growing conditions, harvesting time, storage conditions, type of solvent used, extraction method, and the methods of the analyses [42]. In addition, there were few studies carried out with the same test conditions, mainly for antioxidant activity, which made it difficult to compare the results, especially for laurel extracts.

### 3.3. Antimicrobial Activity of the Extracts

For LLE, MIC values were achieved at 1% for *S. aureus* and *L. monocytogenes,* while for *E. faecalis* it was found to be 1.8%. When OLE was combined with LLE (1:1), the MIC value was achieved at 1% for *S. aureus* and 2% for *L. monocytogenes*. However, for other microorganisms, MIC was not achieved, which indicated that it is higher than 2%. Despite this, it was possible to observe that the absorbance decreased for higher extract concentrations, especially for *Salmonella*. LLE at 2% presented absorbance values below 0.1 (0.06), and comparatively, at 1% and 0.2%, the absorbances found were 0.48 and 0.73, respectively, indicating the sensitivity of the microorganism to concentrations is close to 2%, and a concentration-response. For OLE, MIC was not achieved for any of the microorganisms in the concentration range tested.

For LLE, MBC values were achieved at 2% for *S. aureus* and *L. monocytogenes*. For LLE + OLE, MBC was achieved at 2% for *S. aureus*.

Regarding hydroalcoholic extracts obtained by ultrasound, Fernández et al. [43] determined MIC values between 0.4 and 0.8 mg/mL for *Paenibacillus larvae* (Gram-positive). Mekinić et al. [44], using solid–liquid extraction, obtained a MIC value of 3.33 mg/mL for *E. coli*, 3.33 mg/mL for *Salmonella* Infantis, 1.39 mg/mL for *S. aureus*, and 3.33 mg/mL for *L. monocytogenes*.

MIC and MBC results showed that Gram-positive bacteria were more sensitive than Gram-negative bacteria. The high sensitivity of Gram-positive bacteria can be attributed to the absence of lipopolysaccharides in their cell walls, different to Gram-negative bacteria [45]. The Gram-negative bacteria cell structure includes an outer membrane with a hydrophilic surface rich in lipopolysaccharides that forms an additional protective layer surrounding the cell. This barrier can effectively prevent the entry of active compounds [45,46]. The relative resistance of Gram-negative bacteria can also be attributed to the periplasmic space. The periplasmic space, located between the inner and outer membranes, contains enzymes and proteins that can help neutralise or modify external compounds [47]. The characteristics of microorganisms, plant materials, and chemical properties are important to understanding the relative resistance and antimicrobial efficiency.

Polyphenols are the main plant secondary metabolites present in plant extracts [48]. The main mechanisms of action of polyphenols proposed are the modification of the membrane permeability, formation of cytoplasmic granules, the disruption of the cytoplasmic membrane, changes in intracellular functions induced by hydrogen bonding between phenolic compounds and enzymes through their OH groups, and the modification of fungal morphology, including changes in cell wall rigidity and integrity losses, caused by distinct interactions with cell membranes [49,50].

### 3.4. Antimicrobial Activity of Films

Table 2 presents the counts for the tested microorganisms after 24 h of incubation with the addition of the different films and the control (BHI only, without films). For a better understanding of the count evolution, the results for BHI at 0 h were included.

No antimicrobial activity was observed for the alginate-based film without extracts compared to the control. With the addition of extracts, lower counts were observed at 24 h compared to the control, mainly for Gram-positive microorganisms. This is indicative of a greater difficulty for microbial growth in the presence of these compounds.

After 24 h, and compared to BHI at 0 h, the best results were observed for SA + LLE 1% + OLE 1% against *S. aureus* with a reduction of 1.95 log CFU/g which demonstrates a good antimicrobial activity, although not enough to consider the compound bactericidal (≥3 log10 reduction) [51]. Beyond this, SA + LLE 2% and SA + LLE 0.5% + OLE 0.5% also demonstrated good results against *S. aureus* with a reduction of 1.01 and 1.32 CFU/g, respectively. The application of SA + LLE 1% also inhibited the exponential development of *S. aureus*.

The application of SA + LLE 1% and SA + LLE 2% films also demonstrated the ability to reduce *L. monocytogenes* counts by 0.93 and 1.74 log CFU/g, respectively. The use of SA + LLE 2% also allowed the control of *E. faecalis* development with the counts stabilising after 24 h and when compared with the BHI at 0 h.

The application of films with OLE at 1 and 2% had no effect on controlling microbial growth. However, it was possible to observe lower counts compared to BHI at 24 h. The best results were observed for *S. aureus* applying SA + OLE 1%, with counts approximately 1.5 log CFU/g lower than BHI at 24 h. Similar results were observed with the application of SA + LLE + OLE films against *L. monocytogenes*, *E. faecalis*, and *S.* Typhimurium. The application of SA + LLE 1% and SA + LLE 2% films showed 0.92 and 1.46 log CFU/g counts lower than BHI at 24 h against *S.* Typhimurium.

### 3.5. Migration of Bioactive Compounds

To evaluate the stability and possible inhibition of lipid oxidation, the migration of phenolic compounds and the antioxidant capacity were evaluated for 15 days in water, 10% ethanol, and 95% ethanol. The films solubilised completely after 24 h in water and 10% ethanol. Thus, for these two simulants, only the values after 1 day were recorded (Figure 1 and Appendix A). The results for 95% ethanol are represented in Figure 2 and Appendix A.

The maximum value for phenolic compounds after 24 h was observed for SA + LLE 2% in water (1028.5 mg/L) followed by SA + LLE 1% + OLE 1% in 10% ethanol (898.4 mg/L) and water (876.7 mg/L).

The maximum value for antioxidant activity registered after 24 h was for SA + LLE 2% in water with 8.07 mmol TE/L followed by SA + LLE 2% in 10% ethanol (5.2 mmol TE/L) and SA + LLE 1% + OLE 1% in 10% ethanol (5.2 mmol TE/L). We observed a positive correlation between the total phenolic content incorporated into the films and their antioxidant activity, i.e., films with higher content in phenolic compounds presented better antioxidant properties. For films with LLE at 1 and 2%, the phenolic compounds and antioxidant activity were higher in water. For films with OLE at 1%, the phenolic compounds and antioxidant activity were higher in 10% ethanol, and at 2%, were higher in water. The films with LLE + OLE presented higher phenolic compounds and antioxidant activity in 10% ethanol. The fact that some maximum values were detected for 10% ethanol could be related to some compounds with a greater affinity for ethanol. However, no significant differences were detected between the averages compared to using water as a simulant. The hydrophilic character of the extracts might also explain the results obtained for 95% ethanol after 24 h since lower phenolic compounds and antioxidant activity were detected for all tested films.

The results suggest that the compounds present in LLE and OLE extracts had a higher affinity with the polymeric matrix than 95% ethanol (fatty food simulant), resulting in lower migration. Also, we observed that higher concentrations of extract lead to a faster release rate.

The antioxidant capacity for OLE films in 95% ethanol increased over time (*p* < 0.001) and was even higher on day 15 than the maximum value obtained at 24 h, compared to the other simulants used. The phenolic compounds and antioxidant activity released for films with LLE + OLE also increased over time using 95% ethanol. This increase was less noticeable for the films with LLE, where the antioxidant activity remained more stable over the 15 days, which indicated a low release control. Cui et al. [52] performed a study on the release and antioxidant activity of active potato starch packaging films encapsulating thymol. The authors reported different antioxidant activities in different food simulants due to the varying degrees of solubility of the released thymol. The highest antioxidant activity was in 50% ethanol. Also, the authors reported a similar trend in each simulation; an initial rapid release followed by a sustained release. This trend was observed in our study, mainly for films with OLE and LLE + OLE.

### 3.6. Physico-Chemical Properties of the Films

The produced films were visually homogenous, and uniform, and were easily removed from the petri plate and handled before analysis (Figure 3).

Table 3 shows the effect of LLE and OLE incorporation on moisture content, WVP, *L**, *a**, *b** and opacity of alginate-based films. A decrease in the MC and WVP values and opacity increase in films with extracts can be highlighted.

The water affinity is highly influenced by the addition of extract. Moisture content (MC) values were significantly different between SA films without and with extracts. Results showed that the increase in extract concentration led to lower values of MC. The lowest value observed was for SA + LLE 2% films with 15.07%. This may be explained by the increase of phenolic compounds with hydrophobic characteristics (even in low amounts) and their interaction with alginate. However, adding the extracts does not impact solubility since all the films are solubilised completely. Liu et al. [53] also reported a decrease in moisture content with the addition of pomegranate flesh and peel extracts to the κ-carrageenan matrix. The authors attributed the result to the interaction of polyphenol compounds with κ-carrageenan by the formation of new hydrogen bonds leading to reduced available hydroxyl groups for interaction with water molecules.

The WVP values ranged from 3.49 × 10^−11^ (SA + OLE 1% + LLE 1%) to 87.7 × 10^−11^ g m^−1^ s^−1^ Pa^−1^ (PLA), being significantly different from each other. WVP values of SA films with and without extracts were lower when compared to PLA films. Costa et al. [6], using the same concentration of alginate with different ratios of β-D-mannuronic acid (M blocks) and a-L-guluronic acid (G blocks) and same concentration of glycerol, obtained lower values for SA films than our study (10.5 × 10^−11^ g m^−1^ s^−1^ Pa^−1^ for sodium alginate 65/35 M/G, and 7.83 × 10^−11^ g m^−1^ s^−1^ Pa^−1^ for sodium alginate 30/70 M/G). Alves et al. [54] even found lower values using the same concentrations (5.15 × 10^−11^ g m^−1^ s^−1^ Pa^−1^ for sodium alginate 65/35 M/G). As reported by Costa et al. [6] the molecular weight and sequence of the M and G residues affect the films’ physicochemical properties. The different methodologies used may also have contributed to the different results. The WVP values decreased with extract addition; the higher the extract concentration, the lower the WVP values. The incorporation of higher ratios of hydrophobic compounds may explain these differences. The decrease in WVP values (low permeability values) can be used to increase food preservation [55]. Martiny et al. [56] reported 4.17 × 10^−11^ and 2.26 × 10^−11^ g m^−1^ s^−1^ Pa^−1^ for carrageenan films without and with OLE extract, respectively, demonstrating an improvement with OLE extract incorporation. Different results were obtained by Albertos et al. [57] who reported a significant increase in WVP values with the addition of OLE in gelatin films. The authors obtained 0.73 mm/kPA·h·m^2^ for gelatin films and 1.44 g mm/kPA·h·m^2^ for gelatin films with OLE 5.63% (*w*/*w*). The increase in thickness values was pointed out by the authors as a possible reason for the increase in WVP. Also, the hydrophilic/hydrophobic ratio may explain the differences in the results obtained in the different studies.

The films without extract appeared clear, with *L** values close to 100, associated with the high transparency of the films. The LLE, OLE, and LLE + OLE films showed a yellowish colour, as demonstrated in Figure 3. The low *L** and high *b** values showed a decrease in transparency and an increase in the yellower appearance of films with extracts when compared with SA films. SA + OLE 2% films were 29.1% darker and 94.5% yellower than SA films. The natural colour of the extracts resulted in higher values of Δ*E*. As expected, as the concentration of extracts increased, Δ*E* also increased. SA films with the addition of LLE 2% had a higher Δ*E.* The results also showed that the film’s opacity increased significantly with extract addition, essentially due to the natural colour of extracts. SA films showed lower opacity, 4.33%, and SA + LLE 2% films the highest opacity, 16.43%. The variations for *L** and *b** values were in agreement with other studies [56,57].

### 3.7. Mechanical Properties

The extract incorporation in films affected the thickness, tensile strength (TS), elongation at break (EB), and modulus of elasticity (ME), represented in Table 4. Thickness values ranged from 63.67 (SA films) to 137.37 µm (SA + OLE 2%). The addition of the extracts caused an increase in the thickness of the films. This increase is in agreement with results presented by Martiny et al. [56] who reported 16 µm for carrageenan film with an increase to 58 µm for carrageenan film with olive leaf extract, which is coherent with the greater solid content per surface unit.

The most evident changes in TS were observed between SA and SA + LLE 1% + OLE 1% films, with a decrease of 65.7%. Compared to SA films, the incorporation of OLE 1%, OLE 2%, and LLE 1% showed a decrease (*p* < 0.05) of about 17.7%, 32.5%, and 28%, respectively. Although these values were not different from each other (*p* > 0.05). The reduction in TS as the extract concentration increased may be attributed to the molecular interactions between the film-forming constituents and extract incorporation [58]. Also, the reduction in TS may be a result of the poor dispersion caused by the increase in extract concentration [59].

An improvement of EB was observed for OLE 2% films, with an increase of 24.8%. The decrease in TS and improvement of EB for OLE films may suggest an improvement in flexibility compared to SA and LLE films.

A significant improvement in ME was observed in SA + LLE 1% (54.8%) and SA + LLE 0.5% + OLE 0.5% films (86.7%) showing the possibility of laurel interference in this characteristic of the films (stiffer and less flexible). On the other hand, the presence of olive extract appears to have the opposite effect.

The results are corroborated by the study of Bhatia et al. [58]. The authors reported a decrease in TS and an improvement in EB for alginate films incorporated with Ficus fruit extract. The improvement of EB for films with OLE and ME for films with LLE may be explained by the plasticising effect of the extracts. Several authors reported plasticising properties with the incorporation of natural plant-based compounds [60].

### 3.8. Fourier Transform Infrared Spectroscopy (FTIR)

The FTIR spectra of OLE and LLE extracts and SA films containing LLE and OLE are shown in Figure 4.

The FTIR spectra for LLE and OLE exhibited the presence of many functional groups such as alcohols, phenols, and carboxylic acids, indicated by the bands between 3200 and 3500 cm^−1^, also corresponding to intermolecular hydrogen bonds and O-H stretching vibrations [61,62]. The spectral band observed at 2925 cm^−1^ (OLE) and 2921 cm^−1^ (LLE) is a characteristic feature of C-H stretching vibrations attributed to the presence of aliphatic C-H groups [62]. Peaks in the range of 1550 to 1750 cm^−1^ correspond to the presence of C-O bonds found in esters or carboxylic acids and their derivatives, and C=O stretching of the carboxylic acid with an intermolecular hydrogen bond [61,63]. Peaks within the range of 1000 to 1300 cm^−1^ indicate the presence of alcohols, phenols, aliphatic ethers, and esters [61].

Pure alginate film exhibited five characteristic absorption bands at around 3251 cm^−1^, 2933 cm^−1^, 1598 cm^−1^, 1407 cm^−1^, and 1025 cm^−1^. The broad band at 3251 cm^−1^ results from the stretching vibration of the OH group, which participates in hydrogen bond formation [10,64]. The absorption peak observed at 2933 cm^−1^ is attributed to the stretching vibration of C-H bonds [10]. The peaks observed at 1598 cm^−1^ and 1407 cm^−1^ correspond to the asymmetric and symmetric stretching vibration of the COO groups, respectively [10,64]. Additionally, an observed band at 1025 cm^−1^ indicates the stretching of the C–O–C glycosidic bond [64]. The spectra of the films with OLE and OLE displayed similar transmittance peaks to pure alginate film. However, the bands slightly shifted and presented differences in intensity. These differences were more pronounced as the concentration of the extracts increased, suggesting a possible interaction between the film matrix (alginate) and the extracts. In addition, one peak around 1490 cm^−1^ appeared suggesting the possibility of the extracts interacting with the matrix. The establishment of hydrogen bonds between polyphenols and alginate may explain the changes in the peak position and intensity [65]. Such interactions could improve the physical and mechanical properties of the films. The results are in agreement with other studies that used the same matrix but different active compounds such as guava leaf extracts, *Stryphnodendron adstringens* extract (Bartimão), and Roselle (*Hibiscus sabdarifa* L.) extract [59,66,67].

### 3.9. Scanning Electron Microscopy (SEM)

Figure 5 shows the scanning electron microscopy (SEM) photographs of SA films with and without extract incorporation. The images showed differences between SA films and films with extracts. SA films exhibited a uniform surface with some visible cracks. When LLE and OLE were added, the films’ morphology changed, depending on the concentration of extract. OLE 1% films were rougher and irregularly distributed, while OLE 2% films formed a more uniform structure. LLE films showed some granular vesicles with spherical shapes and smooth surfaces that were free of visible cracks at 1%. At 2%, the films were more uniform and formed a homogeneous network. The mixture of LLE and OLE extracts resulted in a linear and smooth structure with greater uniformity than LLE or OLE films. However, some spherical granular vesicles were also visible without pores or cracks. The results were consistent with the visual and texture analyses (Figure 3 and Table 4). The SA films appeared with more cracks, being less flexible than films with extracts, particularly at 2%.

As mentioned before, the plasticising effects related to the addition of plasticisers and even plant extracts can lead to some improvements in the flexibility and elongation of the film. This allows the accommodation of external stresses and deformations, reducing the likelihood of crack formation [60]. The reduction of the crack formation is also consistent with the mechanical properties results due to the flexibility improving particularly with OLE addition enhancing its plasticising capacity.

## 4. Conclusions

LLE and OLE have demonstrated their potential to be successfully incorporated in alginate-based films acting as antimicrobial and antioxidant materials, which can be used to extend food shelf-life. LLE extracts and films demonstrated promising results for antioxidant and antimicrobial activity. The incorporation of LLE 2% was able to reduce the populations of Gram-positive bacteria such as *L. monocytogenes* and *S. aureus*. Nevertheless, it was also possible to observe a bacteriostatic effect at 1% for *L. monocytogenes* and *S. aureus*.

The incorporation of LLE and OLE changed the alginate-based film’s physico-chemical properties. Regarding mechanical properties, OLE films were more ductile but less elastic and resistant than SA films, while films with LLE were more elastic but less ductile and resistant. The incorporation of the extracts in the films reduced the moisture content and WVP values, which can be related to the hydrophobic character of the extracts. The selection of materials with low WVP can play a critical role in preserving the quality of food products by reducing weight loss during storage. The films were also affected in terms of colour and transparency, but the increase in the opacity values can help reduce the passage of light and thus be useful to avoid food oxidation.

Based on the migration and solubility studies, the use of alginate films with LLE and/or OLE extracts is advisable for foods with lower water content and potential lipid degradation. For foods with a higher water content, the use of coatings instead of films should be considered due to the high solubility of films in water.

Further studies in real foods are needed to optimise the amount of extracts needed to achieve desirable antimicrobial and antioxidant activity that does not compromise the product’s organoleptic properties.

## Figures and Tables

**Figure 1 foods-12-04076-f001:**
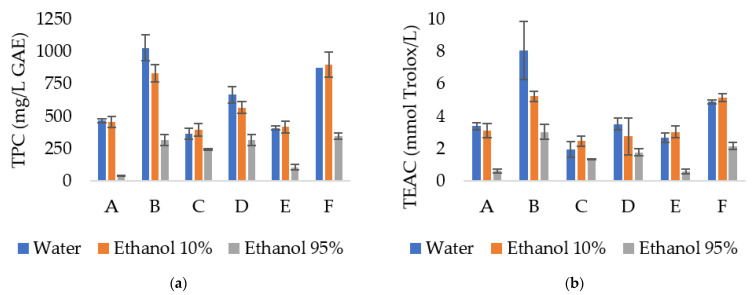
(**a**) Total phenolic compounds (TPC) on simulant media expressed in mg/L gallic acid equivalent. (**b**) Antioxidant activity in simulant media expressed in mmol Trolox equivalent/L simulant. SA—sodium alginate films; OLE—olive leaves extract; LLE—laurel leaves extract; A—SA + LLE 1%, B—SA + LLE 2%, C—SA + OLE 1%, D—SA + OLE 2%, E—SA + LLE 0.5% + OLE 0.5%, F—SA + LLE 1% + OLE 1%.

**Figure 2 foods-12-04076-f002:**
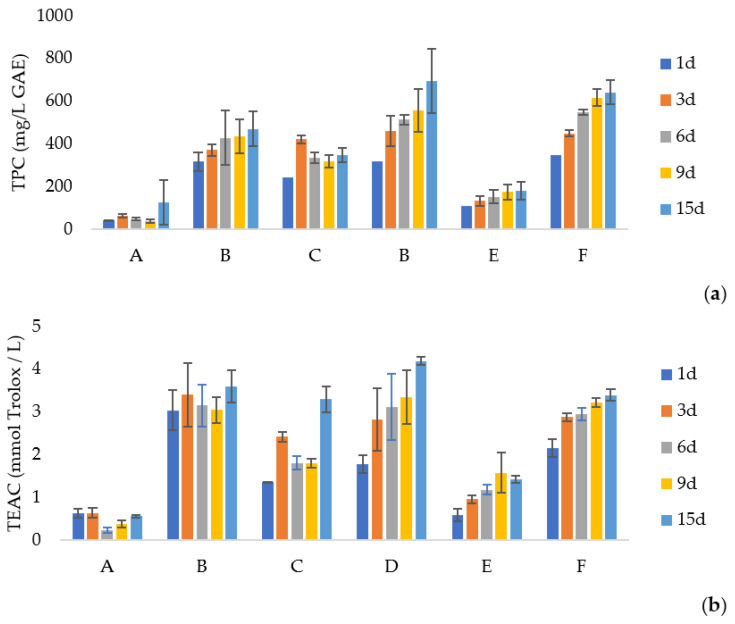
(**a**)—Total phenolic compounds (TPC) expressed in mg/L gallic acid equivalent for 95% ethanol simulant for 15 days. (**b**)—Antioxidant activity expressed in mmol Trolox equivalent/L simulant for 95% ethanol simulant for 15 days. SA—sodium alginate films; OLE—olive leaves extract; LLE—laurel leaves extract; A—SA + LLE 1%, B—SA + LLE 2%, C—SA + OLE 1%, D–SA + OLE 2%, E—SA + LLE 0.5% + OLE 0.5%, F—SA + LLE 1% + OLE 1%.

**Figure 3 foods-12-04076-f003:**
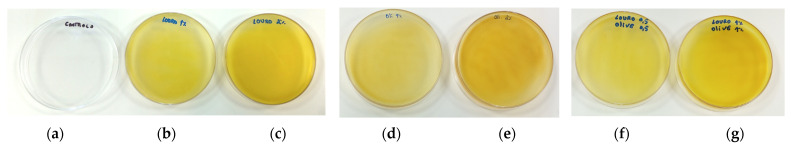
Alginate-based films blended with olive leaf and laurel leaf extracts. (**a**)—SA, (**b**)—SA + LLE 1%, (**c**)—SA + LLE 2%, (**d**)—SA + OLE 1%, (**e**)—SA + OLE 2%, (**f**)—SA + LLE 0.5% + OLE 0.5%, (**g**)—SA + LLE 1% + OLE 1%. SA—sodium alginate films; OLE—olive leaves extract; LLE—laurel leaves extract.

**Figure 4 foods-12-04076-f004:**
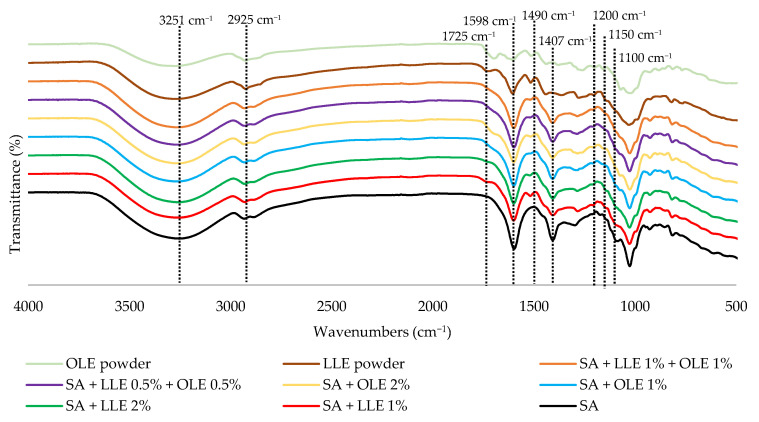
FTIR spectra of extracts, SA films, and SA films with OLE and LLE incorporation. SA—sodium alginate films; OLE—olive leaves extract; LLE—laurel leaves extract.

**Figure 5 foods-12-04076-f005:**
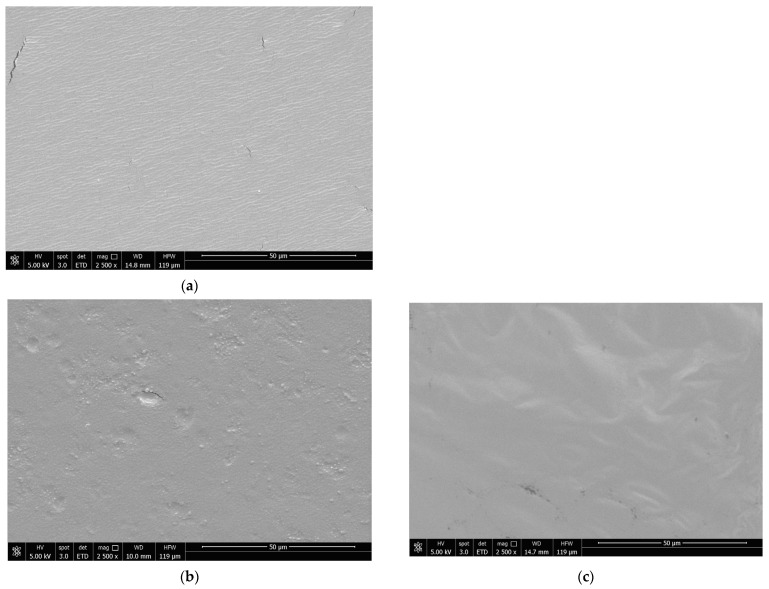
SEM surface images of SA films incorporated with OLE and LLE. (**a**)—SA; (**b**)—SA + OLE 1%; (**c**)—SA + OLE 2%; (**d**)—SA + LLE 1%; (**e**)—SA + LLE 2%; (**f**)—SA + OLE 0.5% + LLE 0.5%; (**g**)—SA + OLE 1% + LLE 1%. Magnification 2500 × and scale bar 50 μm. SA—sodium alginate films; OLE—olive leaves extract; LLE—laurel leaves extract.

**Table 1 foods-12-04076-t001:** Total phenolic compounds and antioxidant activity of extracts and BHT.

Sample	TPC (mg GAE/g Extract)	TEAC (mmol TE/g Extract)	IC_50_ (mg/mL)
LLE	195.75 ± 10.90 ^a^	2.07 ± 0.21 ^a^	0.74 ± 0.00 ^c^
OLE	147.04 ± 1.80 ^b^	0.78 ± 0.02 ^c^	1.77 ± 0.08 ^a^
LLE + OLE	179.13 ± 2.81 ^a^	1.13 ± 0.06 ^c^	0.98 ± 0.03 ^b^
BHT	nd	1.53 ± 0.16 ^b^	0.56 ± 0.04 ^d^
*p*	<0.01	<0.001	<0.001

LLE—laurel leaves extract; OLE—olive leaves extract; BHT—butylated hydroxytoluene; nd—non-determined. Means with different letters (columns) differ significantly, *p* < 0.05.

**Table 2 foods-12-04076-t002:** Counts for tested microorganisms (mean ± standard deviation, log CFU/g) after 24 h of incubation.

Sample	*S. aureus*	*L. monocytogenes*	*E. faecalis*	*S.* Typhimurium	*E. coli*
BHI 0 h	7.26 ± 0.06 ^c^	7.32 ± 0.03 ^c^	7.19 ± 0.05 ^d^	7.09 ± 0.04 ^c^	7.25 ± 0.08 ^f^
BHI 24 h	10.66 ± 0.24 ^a^	10.48 ± 0.19 ^a^	10.17 ± 0.09 ^a^	10.68 ± 0.05 ^a^	12.05 ± 0.02 ^a^
SA	10.70 ± 0.06 ^a^	9.93 ± 0.13 ^ab^	10.02 ± 0.14 ^ab^	10.77 ± 0.03 ^a^	12.05 ± 0.03 ^a^
SA + LLE 1%	7.50 ± 0.25 ^c^	6.39 ± 0.52 ^cd^	9.28 ± 0.37 ^bc^	9.76 ± 0.14 ^b^	11.25 ± 0.15 ^d^
SA + LLE 2%	6.31 ± 0.23 ^d^	5.58 ± 0.66 ^d^	7.41 ± 0.31 ^d^	9.22 ± 0.46 ^b^	11.03 ± 0.10 ^e^
SA + OLE 1%	9.20 ± 0.14 ^b^	9.96 ± 0.17 ^ab^	9.85 ± 0.02 ^abc^	9.86 ± 0.01 ^b^	11.59 ± 0.17 ^b^
SA + OLE 2%	9.35 ± 0.09 ^b^	10.02 ± 0.06 ^ab^	9.52 ± 0.24 ^abc^	9.69 ± 0.20 ^b^	11.37 ± 0.05 ^cd^
SA + LLE 0.5% + OLE 0.5%	6.00 ± 0.26 ^de^	9.55 ± 0.27 ^ab^	9.65 ± 0.02 ^abc^	9.51 ± 0.06 ^b^	11.45 ± 0.11 ^bc^
SA + LLE 1% + OLE 1%	5.31 ± 0.05 ^e^	8.84 ± 0.19 ^b^	9.03 ± 0.33 ^c^	9.35 ± 0.04 ^b^	11.20 ± 0.05 ^de^
*p*	<0.001	<0.001	<0.001	<0.001	<0.001

BHI—Brain heart infusion broth, without film; SA—sodium alginate films; LLE—laurel leaves extract; OLE—olive leaves extract. Means with different letters (columns) differ significantly, *p* < 0.05.

**Table 3 foods-12-04076-t003:** Moisture content, WVP, *L**, *a**, *b**, Δ*E* and opacity of alginate-based films blended with LLE and OLE extracts.

Film	Moisture Content (%)	WVP × 10^−11^ (g m^−1^ s^−1^ Pa^−1^)	*L**	*a**	*b**	Δ*E*	Opacity (%)
PLA	nd	87.7 ± 0.10 ^a^	96.77 ± 0.07 ^a^	0.19 ± 0.02 ^c^	2.01 ± 0.07 ^e^	0.87 ± 0.09 ^f^	9.28 ± 0.10 ^e^
SA	31.55 ± 3.82 ^a^	22.08 ± 1.13 ^a^	96.50 ± 0.38 ^a^	0.16 ± 0.06 ^c^	3.20 ± 0.37 ^e^	1.84 ± 0.47 ^f^	4.33 ± 0.93 ^f^
SA + OLE 1%	23.69 ± 2.23 ^b^	6.49 ± 0.27 ^b^	77.14 ± 2.92 ^d^	4.24 ± 1.71 ^b^	46.20 ± 1.73 ^d^	49.22 ± 1.92 ^e^	11.74 ± 3.12 ^d^
SA + OLE 2%	19.36 ± 2.43 ^cd^	4.29 ± 0.16 ^cd^	68.38 ± 4.01 ^f^	9.89 ± 2.01 ^a^	58.12 ± 3.58 ^b^	64.43 ± 2.70 ^c^	13.37 ± 2.17 ^bcd^
SA + LLE 1%	15.59 ± 2.05 ^d^	5.20 ± 0.11 ^bc^	85.92 ± 1.28 ^b^	−5.24 ± 0.31 ^e^	52.60 ± 4.45 ^c^	52.46 ± 4.56 ^d^	11.93 ± 1.45 ^cd^
SA + LLE 2%	15.07 ± 1.97 ^d^	3.53 ± 0.09 ^cd^	73.82 ± 2.10 ^e^	5.00 ± 1.02 ^b^	71.47 ± 1.50 ^a^	73.87 ± 1.41 ^a^	16.43 ± 1.53 ^a^
SA + OLE 0.5% + LLE 0.5%	20.35 ± 3.23 ^bc^	6.08 ± 0.32 ^b^	81.53 ± 2.10 ^c^	1.34 ± 1.56 ^c^	52.80 ± 4.46 ^c^	53.58 ± 4.67 ^d^	14.00 ± 2.17 ^bc^
SA + OLE 1% + LLE 1%	17.47 ± 3.45 ^cd^	3.49 ± 0.11 ^d^	77.87 ± 2.94 ^d^	−2.72 ± 1.24 ^d^	68.92 ± 2.18 ^a^	70.15 ± 2.19 ^b^	14.14 ± 1.65 ^b^
*p*	<0.001	<0.001	<0.001	<0.001	<0.001	<0.001	<0.001

nd—non determined; PLA—Polylactic acid; SA—sodium alginate films; OLE—olive leaves extract; LLE—laurel leaves extract. WVP—Water vapour permeability. Means with different letters (columns) differ significantly, *p* < 0.05.

**Table 4 foods-12-04076-t004:** Thickness and mechanical properties of the alginate films blended with LLE and OLE extracts.

Film	Thickness (µm)	Tensile Strength (MPa)	Elongation at Break (%)	Modulus of Elasticity (Mpa)
PLA	50.00 ± 1.09 ^e^			
SA	63.67 ± 9.36 ^de^	14.27 ± 5.38 ^a^	40.89 ± 7.37 ^b^	17.65 ± 1.31 ^c^
SA + OLE 1%	97.06 ± 8.76 ^bc^	11.75 ± 2.04 ^a^	48.65 ± 3.78 ^ab^	11.44 ± 0.74 ^d^
SA + OLE 2%	137.37 ± 15.55 ^a^	9.63 ± 1.39 ^ab^	54.39 ± 3.94 ^a^	8.00 ± 0.65 ^e^
SA + LLE 1%	83.77 ± 10.96 ^cd^	10.27 ± 2.86 ^a^	29.40 ± 6.23 ^c^	27.32 ± 2.84 ^b^
SA + LLE 2%	119.03 ± 12.43 ^ab^	4.91 ± 1.18 ^c^	25.59 ± 6.04 ^cd^	15.18 ± 1.42 ^c^
SA + OLE 0.5% + LLE 0.5%	90.47 ± 11.55 ^c^	5.46 ± 0.96 ^bc^	16.83 ± 3.65 ^d^	32.97 ± 3.27 ^a^
SA + OLE 1% + LLE 1%	119.37 ± 18.87 ^ab^	4.90 ± 1.37 ^c^	28.86 ± 5.32 ^c^	11.74 ± 1.17 ^d^
*p*	<0.001	<0.001	<0.001	<0.001

PLA—Polylactic acid; SA—sodium alginate films; OLE—olive leaves extract; LLE—laurel leaves extract. Means with different letters (columns) differ significantly, *p* < 0.05.

## Data Availability

The data presented in this study are available on request from the corresponding author.

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
