# Peer review of "Characterization of Sodium Alginate-Based Films Blended with Olive Leaf and Laurel Leaf Extracts Obtained by Ultrasound-Assisted Technology"

_foods, 2023, doi:10.3390/foods12224076_

Round 1

Reviewer 1 Report

Comments and Suggestions for Authors

You can find my comments below for the manuscript entitled “Characterization of sodium alginate-based films blended with olive leaf and laurel leaf 2 extracts obtained by ultrasound-assisted technology”.

Abstract: “Ultrasonic-Assisted Extraction” should be written as lower case letters, unless the authors introduced this as an explanation for abbreviation.

Line 17-20; Please check and revise this sentence, those given … analyses were performed for the films.

Line 24; Please be specific while you are explaining your results. Such as; …x log reduction for S. aureus etc.

Line 25; significant reduction?

The introduction should highlight the novelty of this article and their utmost contribution to the current literature. There are numerous articles with alginate based films incorporated with various types of bioactive compounds.

Line 106; Was this extraction parameters determined according to a former study of authors? If so, please introduce the citation. If not based on earlier study, then how did they select these parameters?

Why the authors did incorporate directly the ethanolic extracts of leaves without encapsulating them? Is it not important to provide a controlled release of these bioactive compounds especially when coating of any food sample with these films during storage?

Line 117; How many replicates of films were prepared?

Line 160-176; Please check the sentences and correct the unnecessary upper case letters.

Line 186; It is better to modify this title for expressing the migration of bioactive compounds, because it reminds the common analysis for assessing the migration of toxic materials from the packaging films.

Line 242-244; The authors should introduce the units of mechanical properties.

Line 264-271; Are those extracts average of different extractions? If so, please introduce the std. dev. or only one extraction was performed?

Line 281-282; Did you check if there were any significant difference between the samples?

Line 304; IC50 or IC50%?

Line 338; Salmonella Infantis, revise this term.

Table 1; check and revise the writing of m.o.’s. Besides, the authors should explain what is control, what is SA and BHI? It was written as BHI as only alginate film, then what is SA and control? Also why did the authors exclude the CNT 0h from the statistical analysis? They can add the 0h m.o. count of the all films.

Check and revise the writing of “S. Typhimurium” throughout the manuscript.

Line 371-388; How many log reduction of m.o. found effective against that specific m.o.? Is there any limit in the standards? If so, please compare your results with the standard values.

It is better to covert Fig. 1. into a table. It will better to compare the results within a table.

Line 409-410; superior between the tested films? Try not to exaggerate your findings. If it is superior than any other sodium alginate-based films, then cite the compared results.

Line 447; You should confirm and correlate these visual findings with mechanical properties and SEM. Try to discuss your overall findings inter-correlating with other film properties.

Line 477-479; In contrary to the literature, OLE or LLE addition improved the films against water vapor permeance. However, since you did not emulsified or encapsulate the lipophilic essential oils how come they uniformly distributed within the film? Please compare you WVP’s with more than one study given in the literature and if possible with only sodium-alginate based films.

Line 510-512; This sentence is contradicting with your discussion given in WVP and migration!

In Fig.4; the LLE and OLE powder has almost similar band intensities at the same wavelength range. So please check your results and compare with previously published studies if they have really overlapping chemical structures.

Line 558; Can you name those active compounds in this sentence?

Line 566-567; Is this discussion valid for OLE and LLE films together? If so, the films did not represent your discussion. Please check your comments given for SEM images.

Conclusions; Based on your current findings and discussion, which food can be coated with these films and why?

Comments on the Quality of English Language

The authors should especially check and revise the scientific terms. Try to avoid long sentences.

Author Response

The authors are grateful to the Editor and the Reviewers for their attentive and detailed remarks which helped to considerably improve our manuscript.

We hope the answers below and modifications introduced in the manuscript are clear and concise enough as required by the Reviewers in order to enable the publication of the manuscript in Foods.

Reviewer 2 Report

Comments and Suggestions for Authors

This work prepared sodium alginate-based films blended with olive leaf and laurel leaf extracts, the methods are well adapted, the data and discussions are both sufficient, I suggest this paper needs minor revisions.

1. It is better if the authors present a table or figure to illustrate the results of antioxidant activity of the extracts.

2. Figure 4, the specific bands can be labled on the figure. 

Author Response

(The authors gave the same response as above.)

Reviewer 3 Report

Comments and Suggestions for Authors

MDPI - Foods

foods-2681280

Characterization of sodium alginate-based films blended with olive leaf and laurel leaf extracts obtained by ultrasound-assisted technology

General comments:

This manuscript reports findings about properties of alginate-based films incorporated with laurel and olive leaves extracts obtained by UAE method. The idea of using natural plant-based preservatives for extending the shelf life of foods is interesting to follow. Evaluating different properties of extracts and fabricated films have been carried out using a number of useful methods. In general, the authors have proven that their methodology to develop bio-nanocomposite films can improve their antioxidant, antimicrobial, and WVP characteristics, which is valuable for enhancing food preservation.

In general, this study covers a number of useful characterization techniques to evaluate the research hypothesis. In my opinion the topic and research subject are interesting and has novelty. However, the authors are encouraged to justify why they did use their extract in free form and there is no encapsulation method to preserve their bio-active materials, extracted bioactive materials in their free from, are sensitive to environmental factors during storage and packaging, also they may adversely influence sensorial properties of foods packed with these films in food packaging applications.

Specific comments:

Abstract:

1-      Abstract should contain the most important findings of the study, especially in a quantitative form. Please include most important results and provide future outlook of this study at the end of abstract.

Introduction:

2-      Introduction does not provide important information about the objectives and research hypothesis and it needs a deep revision considering following aspects:

·         Description of the core novelty of the study

·         More elaboration and similar active components used in formulation of active films incorporating natural antibacterial agents

·         Sufficient literature review about active films produced so far with regards to chitosan/gelatin-based films

·         Description of the objective and hypothesis of the study

·         The authors are encouraged to justify why they did use their extract in free form and there is no encapsulation method to preserve their bio-active materials, extracted bioactive materials in their free from, are sensitive to environmental factors during storage and packaging, also they may adversely influence sensorial properties of foods packed with these films in food packaging applications.

Materials and Methods:

3-      Please provide the chemical composition of your extracts with normal solvent extraction and UAE methos, to provide the difference, which the paper relies on the ovel UAE method.

4-      The thermal properties (TGA-DTG-DSC) of extracts and active films incorporating added materials are missing. The authors are encouraged to add these data.

Results and Discussion

5-      In Figure 3, please change (a) to A, (b) to B, etc. Because in your explanation you have used A, B, C, …

6-      I would like to see Delta E (color difference), Yellowness index and whiteness index in your color results (Table 2) .

7-       

Conclusion

8-      There are too many paragraphs, which can be combine into a coherent story. Conclusion should not repeat the results, rather it is expected to get more general conclusion out od your results.

Author Response

(The authors gave the same response as above.)

Reviewer 4 Report

Comments and Suggestions for Authors

The abstract provides a clear and concise summary of the study, including its objectives, methods, and significant results. However, it would benefit from including specific quantitative data (e.g., actual values for TPC, antimicrobial effectiveness, or changes in film properties) to give readers a more concrete understanding of the findings. Nonetheless, it effectively communicates the potential value of the research in addressing sustainability and food preservation concerns.

The Introduction contains some extraneous information that does not directly contribute to the focus of the study. For example, while it's valuable to provide some background on alginates, the discussion could be more concise. Similarly, extensive descriptions of Laurus nobilis and olive trees might be more detailed than necessary for an introduction. Instead, a more thorough explanation of the UAE process and its advantages over traditional methods would help readers better understand its importance. Moreover, please refer to previous articles (if any) using this combination of compounds and state the innovation in this paper (if any).

In the Results and Discussion section, the authors give an extensive explanation for the fact that Gram-positive bacteria were more sensitive than Gram-negative bacteria. This is a well-known aspect and means nothing concerning the presented materials.

I recommend presenting the data from Table 1 as a chart.

Line 544 please be careful at measuring units.

From the Conclusion section, it is not clear if the obtained films are recommended for foods with low or high water content; please explain.

The overall impression is that the manuscript lacks novelty and the addition of the extracts/essential oil improves in a very limited manner the antimicrobial and antioxidant properties of alginate.

Comments on the Quality of English Language

Minor editing of English language required.

Author Response

(The authors gave the same response as above.)

Round 2

Reviewer 1 Report

Comments and Suggestions for Authors

The authors have still some flaws regarding the necessary use of capital letters. However, they had improved the manuscript regarding the reviewer comments.

Comments on the Quality of English Language

The authors have still some flaws regarding the necessary use of capital letters.

Author Response

Thank you for your comments.

Thank you for your comments. We solve some minor issues detected in this version.  

I'm not sure we've made the improvement you suggested "regarding the necessary use of capital letters". I tried to find anomalies in the text but had difficulty. If you find it necessary, please indicate these anomalies specifically so that I can change them.

Thank you

Reviewer 3 Report

Comments and Suggestions for Authors

The paper can be accepted in after the revisions applied. 

Author Response

Thank you for your comments. We solve some minor issues detected in this version. 

Reviewer 4 Report

Comments and Suggestions for Authors

The authors improved the manuscript and is suitable for publication.

Comments on the Quality of English Language

English is fine or minor issues detected.

Author Response

(The authors gave the same response as above.)
